# Human Amniotic MSC Response in LPS-Stimulated Ascites from Patients with Cirrhosis: FOXO1 Gene and Th17 Activation in Enhanced Antibacterial Activation

**DOI:** 10.3390/ijms25052801

**Published:** 2024-02-28

**Authors:** Mariangela Pampalone, Nicola Cuscino, Gioacchin Iannolo, Giandomenico Amico, Camillo Ricordi, Giampiero Vitale, Claudia Carcione, Salvatore Castelbuono, Simone Dario Scilabra, Claudia Coronnello, Salvatore Gruttadauria, Giada Pietrosi

**Affiliations:** 1Ri.MED Foundation, 90127 Palermo, Italy; gamico@fondazionerimed.com (G.A.); givitale@fondazionerimed.com (G.V.); ccarcione@fondazionerimed.com (C.C.); sdscilabra@fondazionerimed.com (S.D.S.); ccoronnello@fondazionerimed.com (C.C.); 2Department of Laboratory Medicine and Advanced Biotechnologies, IRCCS ISMETT (Istituto Mediterraneo per i Trapianti e Terapie ad Alta Specializzazione), 90127 Palermo, Italy; ncuscino@ismett.edu (N.C.); scastelbuono@ismett.edu (S.C.); 3Cell Transplant Center, Diabetes Research Institute (DRI), University of Miami Miller School of Medicine, 1450 NW 10th Ave, Miami, FL 33136, USA; cricordi@med.miami.edu; 4Department for the Treatment and Study of Abdominal Disease and Abdominal Transplantation, IRCCS ISMETT (Istituto Mediterraneo per i Trapianti e Terapie ad Alta Specializzazione), UPMCI (University of Pittsburgh Medical Center Italy), 90127 Palermo, Italy; sgruttadauria@ismett.edu (S.G.); gpietrosi@ismett.edu (G.P.); 5Department of General Surgery and Medical-Surgical Specialties, University of Catania, 95124 Catania, Italy

**Keywords:** cirrhosis, ascitic fluid, spontaneous bacterial peritonitis, human amnion-derived mesenchymal stromal cells, lipopolysaccharide, placenta

## Abstract

Spontaneous bacterial peritonitis (SBP) is a severe complication in patients with decompensated liver cirrhosis and is commonly treated with broad spectrum antibiotics. However, the rise of antibiotic resistance requires alternative therapeutic strategies. As recently shown, human amnion-derived mesenchymal stem cells (hA-MSCs) are able, in vitro, to promote bacterial clearance and modulate the immune and inflammatory response in SBP. Our results highlight the upregulation of FOXO1, CXCL5, CXCL6, CCL20, and MAPK13 in hA-MSCs as well as the promotion of bacterial clearance, prompting a shift in the immune response toward a Th17 lymphocyte phenotype after 72 h treatment. In this study, we used an in vitro SBP model and employed omics techniques (next-generation sequencing) to investigate the mechanisms by which hA-MSCs modify the crosstalk between immune cells in LPS-stimulated ascitic fluid. We also validated the data obtained via qRT-PCR, cytofluorimetric analysis, and Luminex assay. These findings provide further support to the hope of using hA-MSCs for the prevention and treatment of infective diseases, such as SBP, offering a viable alternative to antibiotic therapy.

## 1. Introduction

Spontaneous bacterial peritonitis (SBP) is a complication of decompensated liver cirrhosis that is caused by a bacterial translocation from the intestine into the peritoneal cavity, resulting in ascites infection [1,2,3,4]. Patients with SBP show an immune dysfunction characterized by both immunosuppression and persistent inflammation [5,6,7,8]. Circulating innate and adaptive immune cells, including those involved in the complement system, are reduced [9,10]. Innate immune cells are stimulated by both pro-inflammatory molecules derived from translocated bacteria (pathogen-associated molecular patterns—PAMPs) and molecules derived from cellular death (DAMPs) by delaying the inflammation status [11]. In this scenario, persistent inflammation and dysregulation of the immune system is responsible for immune paralysis and the inability to clear infections in end-stage liver disease [12,13].

Antibiotic resistance is a major challenge in the management of SBP, and an alternative therapeutic strategy able to immune-modulate and reduce infection susceptibility needs to be found urgently [14,15].

In recent years, regenerative medicine has achieved a significant breakthrough in the search for new therapies for liver diseases [16,17,18,19]. The approach is based on the use of cells, tissues, or biomaterials to repair, replace, or regenerate damaged organ tissues [20,21,22]. Several research groups have demonstrated the effectiveness of human amnion mesenchymal cells (hA-MSCs) in the treatment of various liver pathologies [19,23,24]. These cells have been shown to also have immunomodulatory and anti-inflammatory properties [25,26], promote regeneration of liver tissue [27,28,29], and are promising candidates for cell-based clinical therapy, given their well-known immunomodulatory properties [19,30,31]. Our group has recently shown, in vitro, that by co-culturing hA-MSCs with infected ascites obtained from patients with cirrhosis with refractory ascites, these stem cells exhibit antibacterial properties, modulate inflammatory cytokine pathways [32,33,34], stimulate the production of anti-inflammatory cytokines, and mediate a balanced immune response toward a resolution of the infection [35]. hA-MSCs have been found to promote anti-inflammatory M2 macrophage polarization without affecting the phagocytosis activity of macrophages and NK cells, reduce C3a complement protein and Ficolin-3 concentrations during the course of infection, and significantly decrease the proliferation of common carbapenem-resistant Enterobacterales strains responsible for SBP, including *E. coli* and *Klebsiella pneumonia* [36].

Therefore, hA-MSC-based cell therapy represents a promising avenue for the prevention and treatment of SBP, offering a viable alternative to antibiotic therapy [36].

In our study, we stimulated, in vitro, lipopolysaccharide (LPS) ascitic fluid (AF) from patients with pneumonia in order to mimic a condition of uncomplicated ascites infection. Moreover, we employed omics techniques to investigate the underlying mechanism by which hA-MSCs modulate the immune system. Omics sciences are a rapidly evolving class of disciplines that utilize high-throughput technologies such as genomics, transcriptomics, proteomics, metabolomics, and computational biology to advance research and clarify mechanisms underlying complex diseases [37,38,39]. The study of transcriptome changes and the comparison of gene expression profiles involved in various pathologies have the potential to advance both therapeutics and diagnostics in health care and promote preventive medicine.

We utilized total RNA sequencing (RNA-Seq) using next-generation sequencing (NGS) to assess the transcriptome changes in hA-MSCs after in vitro co-cultures with infected ascites, populated by white blood cells (A-WBCs) obtained from patients with cirrhosis. Our primary aim was to identify the genes that were overexpressed in hA-MSCs upon exposure to LPS-stimulated ascites and that played a crucial role in augmenting the phagocytic activity of A-WBCs and, consequently, reduced bacterial proliferation. Furthermore, we carried out immune profiling of ascites through the assessment of Th1/Th2/Th17 states.

Our results highlight the in vitro upregulation of the transcription factor FOXO1 and various chemokines, including CXCL5, CXCL6, CCL20, and MAPK13 in hA-MSCs, leading to the enhancement of bacterial clearance. Moreover, we observed a shift in the immune response toward a Th17 lymphocyte phenotype, accompanied by a concurrent decrease in the Th1-Th2 lymphocyte component following 72 h of treatment.

## 2. Results

### 2.1. Phenotypic Characteristics and Evaluation of hA-MSCs

Primary cultures of hA-MSCs obtained from three human amniotic membranes of the placenta showed a vitality of more than 80%, and, in order to evaluate their purity, they were subsequently analyzed by flow cytometry. The data showed a positive expression of CD90 (98.83%), CD29 (99.63%), CD13 (98.77%), and CD73 (93.97%), and a negative expression of hematopoietic lineage markers CD34 (0.33%), CD3 (0.07%), CD45 (0.63%), and HLA-DR (0.80%) (Figure 1). The analysis confirmed the quality of our cells for the co-culture in the experimental design, which retained the MSC phenotype.

### 2.2. Ascites Fluid Leukocyte Phenotype

AF can contain a variety of different leukocyte types, including neutrophils, lymphocytes, monocytes, and eosinophils. The relative proportions of these cells can provide clues regarding the underlying cause of ascites.

Immune cells obtained from ascites of four patients with cirrhosis were profiled using a panel of markers to enumerate T cells (CD3+, CD4+, CD8+), B cells (CD19+), monocytes/macrophages (CD14+), natural killer (NK) cells (CD16+CD56+), and neutrophils (CD15+).

The most abundant leukocytes in ascites fluid were lymphocytes (69.65%), with lower numbers of monocytes cells (8.65%) and neutrophils (5%). As shown in Figure 2, the T cell subpopulations displayed a higher percentage of CD3+ (58.10%), while there was a smaller percentage of NK cells (22.77%) and B cells (4.67%) (Figure 2A,B). Our analysis confirmed the presence of lymphocytes and macrophages in the selected samples used for co-culture experiments.

### 2.3. Gene Expression Profiles and Differential Expression in Human Amnion Mesenchymal Stem Cells (hA-MSCs) Cultured Alone and in Co-Culture Pools

In our study, we employed RNA-Seq to investigate the variation in gene expression between hA-MSCs pool samples cultured alone and in co-culture with ascitic fluid from four patients with cirrhosis. We identified the presence of about 17,000 genes in both cultures. Notably, approximately 37.8% of the genes in hA-MSCs alone and 35.4% of the genes in co-culture hA-MSCs exhibited relatively high expression levels (TPM ≥ 10) (Figure 3). Subsequently, we applied a filtering criterion to focus on genes with a log-fold-change (logFC) absolute value greater than 1, resulting in the identification of 792 differentially expressed genes (DEGs). Among these DEGs, 427 were found to be downregulated, while 365 were upregulated in co-cultured hA-MSCs compared to hA-MSCs alone.

The Kyoto Encyclopedia of Genes and Genomes (KEGG) enrichmentanalysis conducted on the 365 upregulated genes identified in co-cultured hA-MSCs demonstrated their involvement in key biological processes, such as in the TNF signaling pathway and IL-17 signaling pathway. Figure 4 provides an overview of the top 10 KEGG pathways ranked by significance (*p*-value) (Figure 4A), along with the genes primarily associated with those pathways (Figure 4B). These findings highlight the potential role of these differentially expressed genes in regulating immune responses and cytokine activities.

To corroborate the results obtained from RNA-Seq, we performed quantitative real-time polymerase chain reaction (qRT-PCR) analysis on a subset of 10 differentially expressed genes (DEGs) on hA-MSC pool samples cultured alone and in co-culture with ascitic fluid from four patients with cirrhosis. Figure 5 illustrates the fold-change values obtained from both the next-generation sequencing (NGS) data and the qRT-PCR results. Interestingly, we observed a similar trend in gene expression changes between the two techniques, which serves to validate our approach and strengthen the reliability of our findings. The agreement between the NGS and qRT-PCR results further supports the robustness of our methodology and the accuracy of the identified differentially expressed genes.

Figure 6 shows the functional association network derived from the STRING tool for the 10 genes. Through functional enrichment analysis of this inferred network, it becomes evident that CCL20, CXCL5, and CXCL6 participate in distinct pathways, exhibiting significant associations. Specifically, these genes enrich the pathways related to chemokine activity (molecular function gene ontology), antimicrobial humoral immune response (mediated by antimicrobial peptides), and chemokine-mediated signaling (biological process gene ontology).

### 2.4. hA-MSCs Co-Culture Shows an Increase in CD4+ Th17 Profile

T cells are a heterogeneous cell population comprising different subsets that exert distinct roles in cell-mediated immunity. T cells are commonly subdivided into three major groups, regulatory T (Treg) cells, helper T (Th) cells, and cytotoxic T (Tc) cells [40,41]. In order to confirm the gene expression of the obtained data, we analyzed the CD4+ T cell differentiation states by using different marker combinations, as described in Section 4.9 of the Section 4, with the aim of evaluating CD4+ T cell state differentiation after hA-MSCs co-culture compared to CD4+ T cells’ control (CD4+ naïve, CD4+ CM, CD4+ EM, CD4+ TEM). We then evaluated activated CD4+ Th cells differentiated into distinct lineages with characteristic patterns [42,43,44,45], as described in Table 1. After 72 h of co-culture with hA-MSCs, an increase in CD4+ T central memory cells of 75% and a reduction in CD4+ T-naïve cells, effector memory, and terminal effector memory of 25%, 52%, and 81.25%, respectively, was found (Figure 7A,B). At the same time, the data showed a reduction in Th1 and Th2 lymphocytic components (23.81% and 65.3%, respectively) with an increase in the percentage of Th17 lymphocytes compared to the control lymphocytic components (466%) (Figure 8A), as also demonstrated in the graph (Figure 8B), which highlights how the presence of hA-MSCs is able to significantly increase the Th17 lymphocyte (** *p*  <  0.01) component and decrease the amount of Th2 lymphocytes (** *p*  <  0.01). However, there was no significant decrease in the Th1 lymphocyte profile. The trend of the lymphocyte profile was also represented by a cytofluorometry panel, as depicted in Figure 8C. These results confirm, using flow cytometry, the gene upregulation identified via NGS.

### 2.5. The Cytokine Composition of AF from Patients with Cirrhosis after Exposure to hA-MSCs Reflects an Environment Promoting a Lymphocytic Th17 Profile

AF is characterized by the presence of inflammatory factors released by immune cells present in ascites. We evaluated the effects of hA-MSCs on cytokine production after in vitro LPS-stimulated AF, as previously described (the analysis was performed on the conditioned media from four different patients in co-culture with ascitic fluid). The cytokines measured were GM-CSF, IFN gamma, IL-1 beta, IL-2, IL-4, IL-5, IL-6, IL-12p70, IL-13, IL-18, TNF alpha, IL-9, IL-10, IL-17A, IL-21, IL-22, IL-23, and IL-27, and those that were statistically significant are represented in Figure 9. The cytokines tested after 72 h of co-culture with hA-MSCs that were significantly changed were IL-17A (*p* < 0.05; *p* < 0.001), IL-23 (*p* < 0.01; *p* < 0.001), IL-10 (*p* < 0.001; *p* < 0.001), IL-9 (*p* < 0.05; *p* < 0.01), IL-2 (*p* < 0.01; *p* < 0.001), TNF-alpha (*p* < 0.01; *p* < 0.001), and IL-18 (*p* < 0.05; *p* < 0.001) compared to the controls (respectively, A-WBCs and hA-MSCs only). These findings show that hA-MSCs are able to induce a release of cytokines that are involved in stimulating Th17 lymphocytes, confirming the previous analysis showing the upregulation of the IL-17 signaling pathway.

## 3. Discussion

SBP occurs in patients with cirrhosis with decompensated liver disease and is caused by the impairment of the hepatic reticulo-endothelial system in clearing translocated bacteria from the intestine to the peritoneal cavity, resulting in infected ascites [46,47,48]. The only effective therapy for SBP is a proper systemic antibiotic treatment and intravenous albumin supplementation. However, response to antibiotic treatments is currently declining because of multidrug resistant organisms (MDROs) [49].

Alternative immune therapies that can decrease bacterial proliferation are under evaluation [35,36,50,51]. Regenerative medicine holds potential for addressing this issue by modulating the expression of specific immunological targets, which can reduce bacterial load and restore homeostasis.

The immune system’s primary function is to eliminate infectious microorganisms that invade the human body [52,53]. However, immune reactions can lead to cellular damage and excessive elimination of targets during chronic inflammatory reactions, leading to autoimmune diseases [54]. Therefore, achieving a balance between the immune system’s pro-inflammatory response to fight bacterial infections and the subsequent production of anti-inflammatory molecules that restore homeostasis after the resolution of the infections is crucial. For this reason, we utilized hA-MSCs to harness both their antibacterial capacity [33,55,56] and their anti-inflammatory/immunomodulatory activity to obtain a resolution of infection, restoration of cytokine production, and immune balance in AF.

In this in vitro study, we employed RNA sequencing (RNA-Seq) technology to investigate how hA-MSCs modulate the phagocytic activity of immune cells and the production of inflammatory mediators during bacterial infections contributing to bacteria clearance in AF. The experimental design involved the co-culture of hA-MSCs with LPS-stimulated ascitic white blood cells (A-WBCs) isolated from four patients with cirrhosis, followed by gene expression profile analysis using RNA-Seq. We then compared the gene expression of co-culture with that individual cell type grown alone, as controls. The differentially expressed gene (DEG) analysis allowed us to identify candidate genes responsible for the observed changes in the immune response. RNA sequencing is an advanced next-generation sequencing technology that can, through a global analysis of RNA transcription, identify and quantify all gene transcripts of a biological sample simultaneously and with a high resolution. To ensure the reliability of our findings, we employed various approaches for gene screening analysis. Through RNA-Seq, we found that 792 genes were deregulated, with 427 of them being downregulated and 365 being upregulated. Furthermore, we utilized KEGG to identify several differentially expressed genes (DEGs) and found differences associated with the TNF signaling pathway (SOCS3; CXCL6; CCL20; TNFAIP3; MAP3K8; CXCL5; IL18R1; MAPK13), beta-alanine metabolism (C3; CXCL6; NOD1; CXCL5; MAPK13), glutathione metabolism (GSTM3; ANPEP; IDH2; MGST1), IL-17 signaling pathway (CXCL6; CCL20; TNFAIP3; CXCL5; MAPK13), TGF-beta signaling pathway (BAMBI; ID2; ID1; ID3; FBN1), glycolysis/gluconeogenesis (ALDH2; ALDH3B1; ALDOC; GAPDH), AGE-RAGE signaling (SERPINE1; PIM1; F3; FOXO1; MAPK13), and viral protein interaction with cytokine and the cytokine receptor pathway (CXCL6;CCL20;CXCL14;CXCL5;IL18R1) (Figure 4). We also found differences in the Th17 and Th1/Th2 cell differentiation pathway. To corroborate our RNA-Seq findings, we employed qRT-PCR and flow cytometric analysis, enabling the identification of a subset of DEGs within the aforementioned pathways. This investigation enabled the characterization of a forkhead box O1 (FOXO1) exhibiting a discernible shift in expression, thus corroborating its substantial role in host response mechanisms, as established in previous studies [57,58,59,60]. In recent years, the transcription factor FOXO1 has gained substantial attention due to its different functions in cellular homeostasis and disease pathogenesis. Beyond its well-established role in metabolism and stress responses, mounting evidence suggests that FOXO1 exerts a crucial influence on the antimicrobial activities of immune cells. In particular, FOXO1 exhibits dynamic regulation and localization in immune cells, orchestrating the transcriptional programs necessary for mounting an effective antimicrobial response [61]. In macrophages, FOXO1 activation promotes the production of pro-inflammatory cytokines and chemokines, facilitating the recruitment and activation of other immune cells at the site of infection, and coordinating the expression of antimicrobial peptides, such as defensins and cathelicidins, contributing to direct antimicrobial defense [62,63,64,65]. The data obtained in this study show that FOXO1 was upregulated in hA-MSCs in contact with LPS-stimulated ascites (Figure 5) and that it may be responsible for the increased expression of target genes that contribute to the activation and regulation of the leukocytes involved in the increase of bacterial phagocytosis.

This study further revealed that the comparison of the hA-MSC transcriptome in co-culture with A-WBCs, as opposed to hA-MSCs grown alone, demonstrated an upregulation of the TNF signaling pathway, IL-17 signaling pathway, and the Th17 cell differentiation pathway. This upregulation was observed through the increased expression of various genes, such as CXCL6, CCL20, CXCL5, and MAPK13 [66,67]. During a response to infection, various inflammatory mediators are produced, including cytokines, chemokines, and growth factors. One of the most widely studied cytokines in the context of bacterial infections is interleukin 17 (IL-17), produced by the Th17 subset of CD4+ T cells [68]. IL-17 plays a critical role in host defense against extracellular bacterial pathogens by stimulating the production of antimicrobial peptides and by recruiting neutrophils and other immune cells to the site of infection. The Th17 subset of CD4+ T cells has emerged as a crucial component of the immune response to microbial infections. The data obtained in this study show how the interaction with hA-MSCs is able to maintain a balanced activation through an increase of cytokine IL-17 and IL-23 (Figure 9A) and simultaneously of the anti-inflammatory cytokine IL-10 and IL-9 (Figure 9B) [69].

In detail, Th17 lymphocytes represent a crucial source of interleukin-17 (IL17) and depend on interleukin-23 (IL-23) to sustain their phenotype and produce high levels of IL17 [70,71,72]. Specifically, IL-17 induces the activation of nuclear factor-kB (NF-kB) and triggers the expression of pro-inflammatory genes in synergy with TNFα, chemokine ligands (CXCL1, CXCL 2, CXCL5, CCL2, and CCL5), and matrix metalloproteinases (MMP3 and MMP13) [73]. On the other hand, cytokines such as IL-4, IL-12, IFN-γ, and IL-27 suppress Th17 differentiation and the expression of Th17 cytokines [74,75]. Cytometric and Luminex assay analyses have revealed a significant decrease in the expression of the Th2 lymphocytic component in the AF after 72 h of co-culturing with hA-MSCs, accompanied by a concomitant increase in the Th17 lymphocytic component (Figure 8), confirmed by the reduction in naïve, effector, and terminal state T cells, as shown in Figure 7. Naïve T cells refer to those that have not encountered their specific antigen and have not yet undergone activation or differentiation. Effector T cells, on the other hand, are T cells that have encountered antigens, undergone activation, and differentiated into specific effector subsets, such as Th1, Th2, or Th17 cells. Memory T cells are a long-lived population of T cells that persist after the resolution of an immune response and can rapidly respond to the re-exposure of the same antigen [76,77]. When naïve T cells differentiate into Th17 cells, there is a decrease in the naïve T cell population. The activation and differentiation processes involve the proliferation and expansion of naïve T cells into effector Th17 cells, leading to a decrease in the pool of naïve T cells [78]. This effect was further confirmed by the measurement of cytokines in the culture medium, which showed a marked increase in the levels of IL-17A, IL-23, IL-9, and IL-10 in the presence of hA-MSCs, while there were no significant changes in the concentration of IL-4, IL-27, and IFN-gamma. Furthermore, we observed an increase in the production of the cytokine IL-2, which plays a crucial role in promoting the growth and expansion of T lymphocytes and TNFα, which are the most abundant early mediators in inflamed tissues responsible for the activation and recruitment of inflammatory cells [79,80] in the presence of hA-MSCs (Figure 9C).

Interestingly, our data revealed that hA-MSCs also increased the expression of IL-18 (Figure 9C), a cytokine responsible for the differentiation of CD4 lymphocytes into Th1/Th2 lymphocytes, which are crucial for cell-mediated immunity against intracellular pathogens through macrophages.

During infection, chemokines play a crucial role in recruiting and activating immune cells to counteract the invading pathogen(s) [81,82,83,84,85]. The data obtained via RNA sequencing have highlighted the ability of hA-MSCs to upregulate some chemokines, such as CXCL6, CXCL5, and CCL20, which are identified as important mediators of Th17 lymphocyte recruitment in infected tissues [86], as demonstrated in the interaction network depicted in Figure 6, and MAPK13 [87].

CXCL6, produced by endothelial cells and macrophages, promotes the attraction of neutrophils and monocytes to the site of infection, facilitating their entry into infected tissues and enhancing the inflammatory response [88]. IL-17A promotes CXCL5, which has a similar effect to CXCL6, and attracts neutrophils, eosinophils, and T cells to the site of infection, enhancing the local immune response [89,90]. Finally, CCL20 chemokine is produced mainly by epithelial cells and macrophages. It attracts specific immune cells, such as regulatory T cells, which play an important role in controlling inflammation [91,92,93]. In summary, CXCL6, CXCL5, and CCL20 chemokines are important for the mechanism of immune signaling during infections, playing a fundamental role in recruiting immune cells to the site of infection and enhancing the local immune response [90]. Their understanding could open up new therapeutic avenues for controlling infections.

Our findings, taken together, suggest that hA-MSCs have a profound immunomodulatory effect on the Th1/Th2/Th17 balance in AF and a demonstrated antibacterial action, potentially offering a strategy to design new therapeutic approaches for the management of immune paralysis and peritoneal infections in end-stage liver failure. Our data showed the mechanism through which hA-MSCs can increase the phagocytic capacity of the immune system, which can be based on an upregulation of gene expression for FOXO1 and the chemokine genes (CXCL6, CXCL5, CCL20) responsible for the activation of the CD4+ Th17 lymphocyte phenotype. This activation is essential for the antibacterial pathways, as described above. Despite our results being derived from four patients, the identification of these upregulated genes and the relative involvement of the lymphocyte pathways responsible for antibacterial actions may be useful in prompting further clinical studies. This study presents a limitation due to the restricted number of patients considered, which can be overcome by recruiting a larger cohort of subjects and performing further in vivo experiments. Obtained data analyses were supported by different methods (NGS, RT PCR, immune profiling) and could be of great interest in finding an alternative to the clinical problems inherent to antibiotic resistance in the treatment of infectious diseases.

## 4. Materials and Methods

### 4.1. Patient Characteristics and Ascitic Fluid Collection

AF was obtained from 4 male patients with cirrhosis mean age 58.5 (range 55–71), in Child C class (according to Child–Pugh score) complicated by refractory ascites (see Table 2.). Liver cirrhosis etiology was related to metabolic dysfunction-associated steatotic liver disease (n = 2), HCV (n = 1) and alcohol (n = 1). Patients were admitted to IRCCS ISMETT (Istituto Mediterraneo per i Trapianti e Terapie ad alta specializzazione) in Palermo, Italy, and underwent standard care with routine paracentesis. Informed consent was obtained from all patients that participated in this study. AF was collected during paracentesis, and signs of infection were excluded (polymorphonuclear leukocyte count in the AF was <250 cells/mm^3^, and AF culture resulted negative after 5 days). An amount of AF ranging from 1 to 2 L was collected from each patient, as explained above. In order to create an in vitro SBP model, LPS was added in all in vitro conditions.

### 4.2. Isolation and Characterization of Human Amnion-Derived Mesenchymal Stromal/Stem Cells

The hA-MSCs were isolated within 6 h after birth from the amnion of human term placentae (n  =  3) from healthy donors at 36–40 weeks of gestation. Mothers’ informed consent was obtained according to the tenets of the Declaration of Helsinki and local ethics regulation (IRRB/58/13, ISMETT Institutional Research Review Board). Isolation was carried out following a well-established protocol, as described in our previous work (35). The expression of hA-MSC markers was analyzed by procuring the isolated cells, washing them twice with FACS buffer (PBS containing 0.3% bovine serum albumin and 0.1% sodium azide), and staining them with antibodies against positive cell surface antigens, CD90 (PE, Clone 5E10, Mouse BALB/c IgG1, κ, BD Biosciences, San Jose, CA, USA), CD73 (APC, Clone AD2, Mouse IgG1, k, Miltenyi Biotec), CD29 (APC, Clone MAR4, Mouse BALB/c IgG1κ BD), and CD13 (APC, Clone WM15, Mouse IgG1 κ BD), and negative cell surface antigens CD45 (PE-Cy7, Clone HI30, Mouse IgG1 κ BD), CD34 (PE-Cy7, Clone 8G12, Mouse BALB/c IgG1 κ BD), CD3 (PerCP-Cy5.5, Clone UCHT1, BD), and HLA-DR (PE, Clone G46-6, BD), for the characterization of MSCs. The cells were then washed twice with the FACS buffer, and the analyses were performed with FACS Celesta SORP flow cytometer and FACS Diva software version 9.0. (BD Biosciences, San Jose, CA, USA).

### 4.3. Leukocyte Subset Analysis of Ascitic Fluid

Ascites immune cells from 4 patients were profiled using a panel of markers to enumerate monocytes/macrophages (CD14+), neutrophils (CD15+), and lymphocytes cells (CD45+). We further phenotyped ascites using a panel of markers that distinguish lymphocyte cell subsets (T cells CD3+, T helper CD4+, T cytotoxic CD8+, B cells CD19+, and NK cells CD16+-56+) using a BD FACS Celesta SORP instrument (Table 3). Analyses were completed using FACS Celesta SORP flow cytometer and FACS Diva software version 9.0 (BD Biosciences, San Jose, CA, USA).

### 4.4. Isolation and Stimulation of White Blood Cell (A-WBC) Component from Ascitic Fluid (AF)

AF was obtained from 4 patients with cirrhosis after paracentesis, and the A-WBCs were obtained after centrifugation of the initial ascites at 300× *g* for 10 min. The red blood cells were lysed through 10 min of incubation with 1× erythrocyte lysis solution (10× Stock Solution: 41.4 g NH_4_Cl, 5 g KHCO_3_, and 1 mL EDTA 0.5 M pH 8 in 500 mL double distilled H_2_O) and subsequent centrifugation at 300× *g* for 10 min. Total A-WBCs were counted with an XN-3000 cell counter (Sysmex Landskrona, Landskrona, Sweden), and the cell pellet was resuspended in an appropriate volume of ascites for each respective patient.

### 4.5. In Vitro hA-MSCs and A-WBC Co-Cultures

Both hA-MSCs and A-WBCs were treated with 0.1 μg/mL of lipopolysaccharide (LPS). All culturing was carried out with a pool of three batches of hA-MSCs and 4 different ascites samples from patients with cirrhosis. In particular, for each patient, only hA-MSCs pooled and A-WBCs grown alone, without co-cultures, were used as controls. While the cell co-cultures were performed using transwells with an insert (12-well, Greiner Bio-One, Kremsmünster, Austria). All conditions were cultured in duplicate for 72 h with LPS, as described above (n = 8).

For each patient, pooled hA-MSCs were plated in four wells at a seeding density of 153,000 cells/well using Chang Medium C and incubated at 37 °C and 5% CO_2_. Two wells represented the first control (hA-MSCs only), while the other wells were subsequently used for co-culture.

The starting time point was established when hA-MSCs reached 80% of confluence. From that time on, Chang Medium C was removed in all wells and replaced with antibiotic-free RPMI in the control wells, and with the respective AF in the co-culture wells. Then, the insert, seeded with 1.12 × 10^5^ WBCs, was placed inside the co-culture wells. The second control with only 3 × 10^5^ A-WBC, resuspended in AF, was seeded in two more wells. Each culture condition was established and incubated for 72 h at 37 °C and 5% CO_2_ with LPS. After 72 h, hA-MSCs were instead tested by different assays in order to evaluate the changes in transcriptome profiles and after co-culture compared to cells grown alone in control wells. We also evaluated the Th1/Th2/Th17 profile after in vitro culture of each condition. All the conditioned media were collected and stored at −80 °C until their use for subsequent analyses to evaluate the secreted cytokines.

### 4.6. RNA-Seq, Library Construction, Sequencing, and Analysis

Total RNA was extracted using an RNeasy Micro Kit, according to the manufacturer’s instructions (QIAGEN, Hilden, Germany), from pooled samples of alone and co-cultured hA-MSCs from the 4 patients. Thus, concentration and quality were assessed with a Qubit 2.0 Fluorometer (Life Technologies, Carlsbad, CA, USA) and 4200 TapeStation System (Agilent Technologies, Santa Clara, CA, USA). Libraries were constructed from 1 µg of RNA extracted from single hA-MSCs and co-culture samples. TruSeq mRNA V2 Sample Preparation Kit with Ribo-Zero Gold (Illumina, San Diego, CA, USA) was employed to generate poly-A-enriched strand-specific libraries. Quality and yield of the obtained libraries were finally determined using a Qubit 2.0 Fluorometer (Life Technologies, Carlsbad, CA, USA) and 4200 TapeStation System (Agilent Technologies, Santa Clara, CA, USA). RNA sequencing was conducted on a NextSeq™ 550 (Illumina, San Diego, CA, USA) with 2 × 76 cycles, in compliance with the manufacturer’s instructions; quality control was completed on the raw sequencing data with FastQC (v0.11.9, Babraham Institute, Babraham, UK), and Trimmomatic (v0.32) was employed to remove low-quality reads and to perform adapt trimming. After this procedure, the extracted RNA was mapped to human reference genome hg19 using STAR (v2.7.0); RSEM (v1.3.3) was used to compute transcript abundances, and gene expression level was normalized by transcripts per kilobase million (TPM). In this study, fold change was computed in order to identify differentially expressed genes (DEGs); specifically, we defined DEGs as those having the logarithm of fold-change (FC) values > 1 or <−1 (|log(FC| > 1). Finally, we analyzed the obtained data in R (v4.1.2) by employing an average linkage algorithm to hierarchically cluster the filtered DEGs, considering the Euclidean distance as the distance metric, and by performing a principal component analysis (PCA).

### 4.7. Real-Time PCR Analysis of Gene Expression by TaqMan Low Density Arrays

To further validate the results obtained by the above-described approach, gene expressions were profiled via TaqMan Gene Expression Arrays according to the manufacturer’s instructions (Thermo Fisher Scientific, Waltham, MA, USA). Total RNA was extracted with miRNeasy Mini Kit and treated with DNAse according to the manufacturer’s instructions (QIAGEN, Hilden, Germany). The purity and quantity of isolated RNA were assessed with OD260/280 with a NanoDrop ND-1000 Spectrophotometer (Thermo Fisher Scientific, Waltham, MA, USA). In total, 1000 ng of RNA was then reverse-transcribed using the high-capacity RNA-to-cDNA kit protocol (Thermo Fisher Scientific, Waltham, MA, USA) to produce single-stranded cDNA. QRT-PCR of 31 human genes was completed with the Applied Biosystems 7900 HT Real-Time PCR system (Waltham, MA, USA). The initial data output for each gene was conducted in SDS software v2.4; all values were expressed as mean ± SD. Finally, we ran a statistical analysis using R functions (v4.1.2).

### 4.8. Pathway Enrichment Analysis

We used the Enrichr Web tool (https://maayanlab.cloud/Enrichr/, accessed on 11 January 2024) to search for the pathways where DEGs are involved, according to the Kyoto Encyclopaedia of Genes and Genomes (KEGG). In Section 2, we listed the top 10 significant ranked terms by *p*-value with histograms and clustergrams. Additionally, we employed the STRING database online tool (v11.0) for the examination of interactions among 10 DEGs. This involved constructing a network representation wherein individual genes were represented as nodes, and the edges denoted the anticipated functional associations between them.

### 4.9. T helper (Th) Cell Profiles after Co-Culture Infection

To evaluate CD4 T-naïve differentiation, the lymphocyte component of the ascites was analyzed in all culture conditions from 3 patients. Specifically, after 72 h of co-culture with hA-MSCs, the maturation status (naïve, central memory, CM; effector memory, EM; terminal effector memory, TEM) of CD4+ cells was analyzed and compared to the control lymphocytes. We phenotyped T cells using a panel of markers that distinguish lymphocyte cell subsets in T helper CD4+ Th1/Th2/Th17. In detail, we analyzed the CD4+ T cells’ differentiation states using different marker combinations, CD4+ naïve (CD45+CCR7+), CD4+ CM (CD45RA-CCR7+), CD4+ EM (CD45RA-CCR7-), and CD4+ TEM (CD45RA+CCR7-), and the lymphocytes cells subsets in T-helper CD4+ using specific marker combinations, Th1 (CD4+CXCR3+CCR4-), Th2 (CD4+CCR4+CCR6-), and Th17 (CD4+CCR4+CCR6+). The cells were labeled with CD45 APC-Cy7, CD3 BV510, CD4 BV786, CD45RA PE Cy-7, CCR7 BV711, CXCR3 BV421, CCR4 APC, and CCR6 BB515—all from BD Biosciences (BB: Brilliant Blue, BV: Brilliant Violet). Analyses were conducted using a BD FACS Celesta SORP instrument, FACS Celesta SORP flow cytometer, and FACS Diva software version 9.0 (BD Biosciences, CA, USA).

### 4.10. Detection of Multiple Cytokines Secreted by Luminex

The levels of different paracrine factors involved were determined in each co-culture-conditioned medium compared to conditioned media secreted by controls (hA-MSCs grown alone in RPMI and A-WBCs grown alone in AF) in triplicate. Cytokine analysis was completed using a Th1/Th2/Th9/Th17/Th22/Treg Cytokine 18-Plex Human ProcartaPlex Panel (Thermo Fisher Scientific, Waltham, MA, USA) based on Luminex technology, according to the manufacturer’s instructions. The data were acquired with xPONENT^®^ 3.1 software for Luminex 100/200 (Luminex Corporation, Austin, TX, USA). The concentration of each factor was calculated via interpolation from standard curves. Results are shown as fold increases of each conditioned medium. ProcartaPlex™ Analyst 1.0 was used for analyzing the data obtained.

### 4.11. Statistics

As already described, all the experiments were performed as four independent experiments using an hA-MSCs pool from three different donors and four different AFs. Data groups were compared using different controls and by applying paired *t*-tests. Results were expressed as the mean ± standard deviation (SD). Differences were considered statistically significant at *p* < 0.05. In the NGS analysis, we conducted experiments by comparing all pooled samples of hA-MSCs cultured alone with all co-cultured pooled samples from four independent experiments (four patients), as described in Section 2.5. Differentially expressed genes (DEGs) were defined based on their fold-change (FC) values, with DEGs identified as those having an absolute log-fold change (|log(FC)|) greater than 1.

## 5. Conclusions

In conclusion, human amnion-derived mesenchymal cells in the presence of an in vitro SBP model, obtained through in vitro stimulation with LPS of AF from decompensated patients with cirrhosis, promoted a robust Th17 immune response. This stimulation led to the recruitment of neutrophils and an upregulation of FOXO1, CXCL6, CXCL5, and CCL20 mediators. As a result, bacteria and bacterial products were more efficiently phagocytosed, leading to a decrease in bacterial load. Simultaneously, the production of anti-inflammatory molecules was amplified, contributing to the restoration of proper immunological balance following the resolution of infection. These findings provide insights into the potential therapeutic manipulation of the Th17 immune response, and offer promising avenues for the management of diseases related to bacterial infections, such as SBP.

On the basis of these results, our future goal will be as follows: to evaluate the miRNAs involved in the increase of phagocytic activity in the same culture conditions, and in presence of hA-MSCs, and to subsequently translate the experiments in vivo.

## Figures and Tables

**Figure 1 ijms-25-02801-f001:**
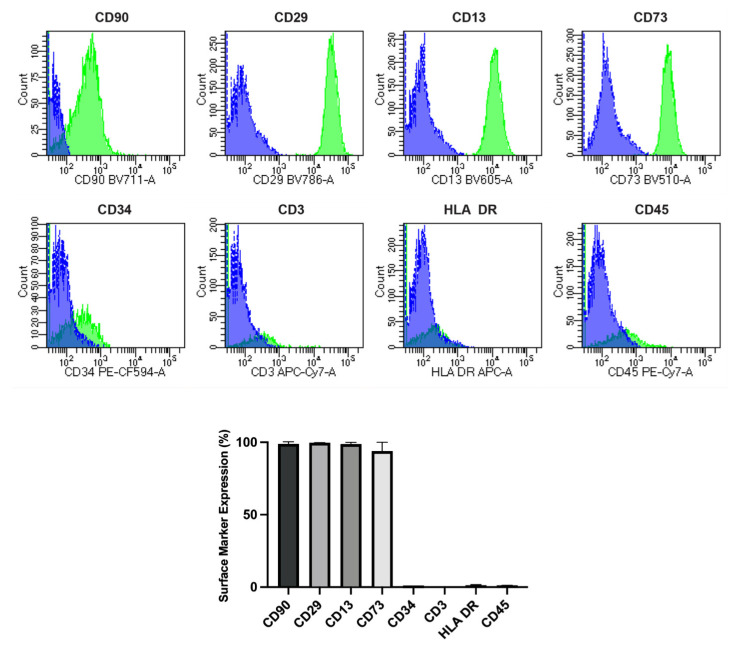
hA-MSC markers. Graph showing expression of positive (CD90, CD29, CD13, and CD73) and negative (CD34, CD3, HLA-DR, and CD45) cell surface markers after human amnion mesenchymal stem cell (hA-MSCs) isolation, with representative images of flow cytometry analysis for the quantification of both positive and negative surface markers (Blue isotype control, green positive staining; the panel is representative of three different analyses).

**Figure 2 ijms-25-02801-f002:**
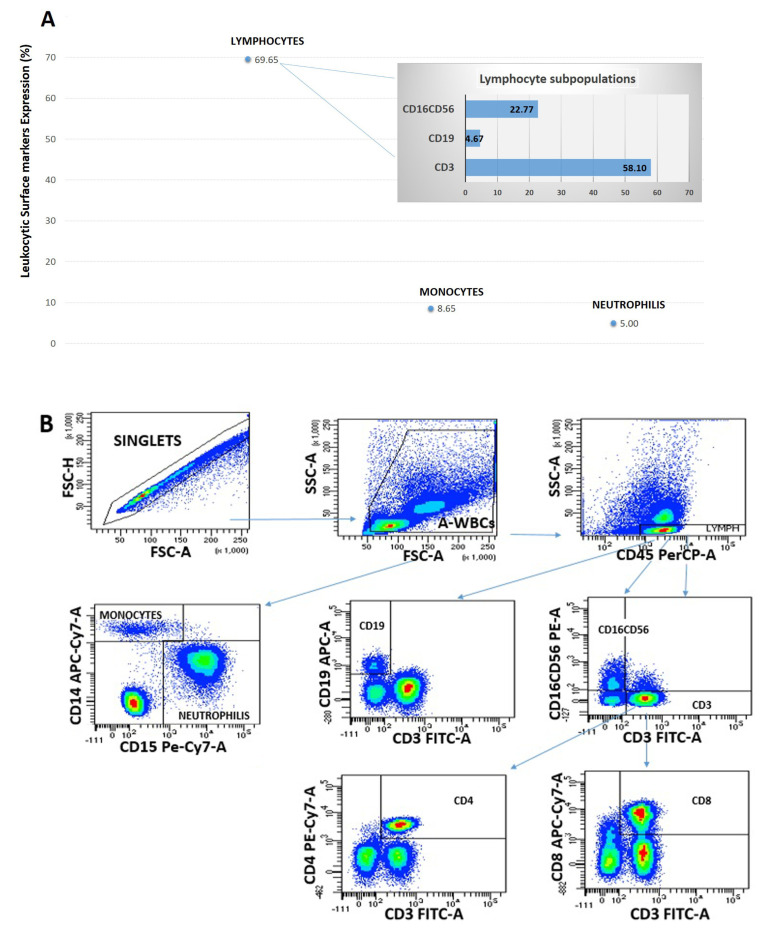
Leukocytes. A representative image showing leucocyte composition from total WBC post-paracentesis AF, in which lymphocyte cells are prevalent compared to monocytes and neutrophil cells. In detail, the lymphocytic component presents a prevalent subpopulation of CD3+ T cells, followed by a smaller percentage of B cells and NK cells, respectively (**A**). Representative flow cytometry analysis (one out of four patients with cirrhosis) for ascites leukocyte characterization (colors represent a density gradient red more dense blue less) (**B**).

**Figure 3 ijms-25-02801-f003:**
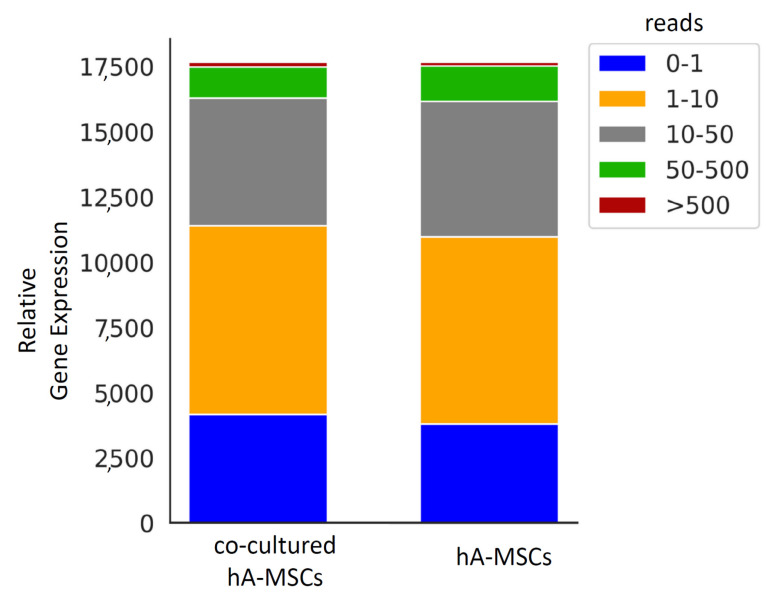
Gene expression profiles and differential expression in human amnion mesenchymal stem cells (hA-MSCs) cultured alone and in co-culture pools. Distribution of gene expression in hA-MSC pools cultured alone and in co-culture.

**Figure 4 ijms-25-02801-f004:**
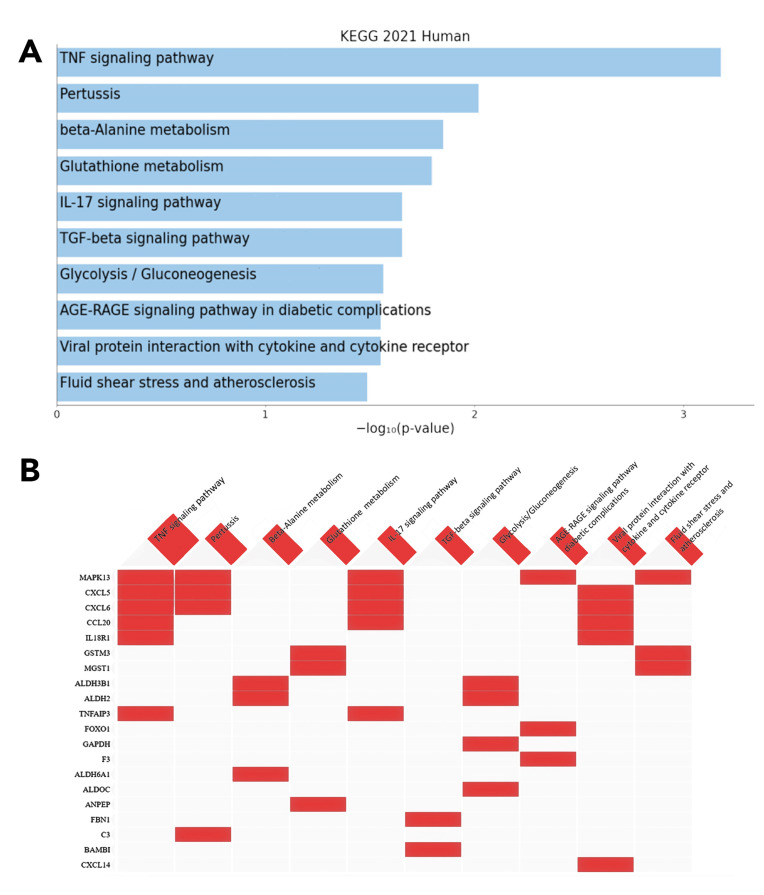
Functional enrichment analysis of KEGG pathway terms from the top 365 upregulated genes in co-culture vs. hAMSCs alone. (**A**) Top 10 significant KEGG functional pathways (*p* < 0.05). (**B**) Clustergram of top 10 ranked KEGG functional pathways showing the top 20 most involved genes.

**Figure 5 ijms-25-02801-f005:**
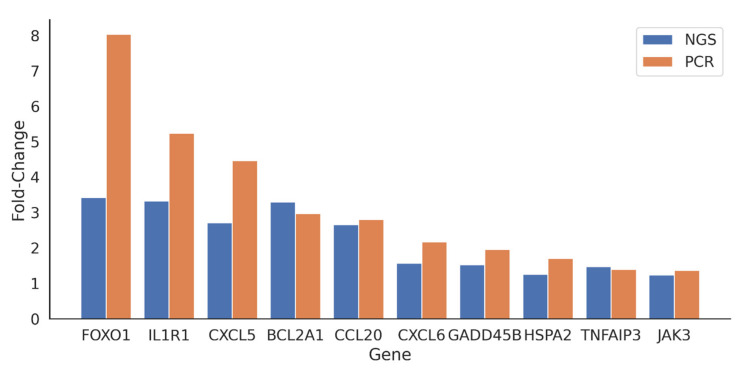
Transcriptomic analysis revealed co-culture-induced hAMSC bioactivity. Upregulated genes were corroborated using qRT-PCR, indicating a similar trend (the analysis was performed on a pool of hA-MSCs alone (ctrl) and in co-culture with ascitic fluids from four patients).

**Figure 6 ijms-25-02801-f006:**
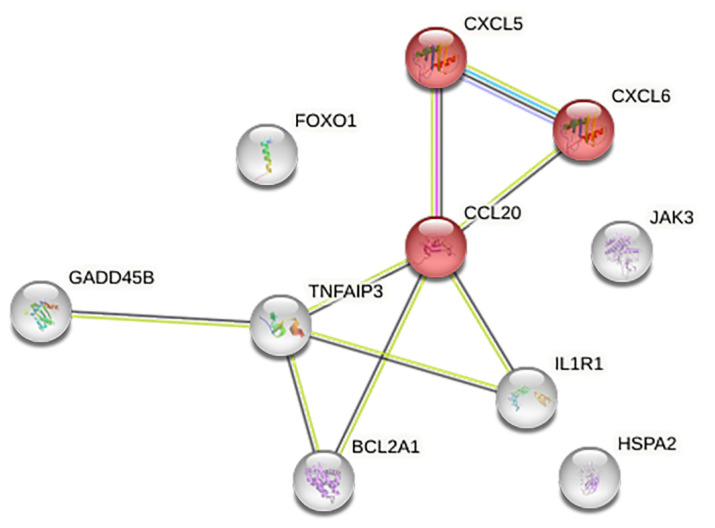
STRING-derived functional network (v11.0) for the 10 differentially expressed genes (DEGs) validated using qRT-PCR. Each node denotes a specific gene, while edges symbolize functional relationships or associations between the genes. Significant genes with major associations are highlighted in red. The data obtained showed an upregulation of chemokines, related to each other, involved in the Th17 lymphocyte pathway, according to KEGG analysis.

**Figure 7 ijms-25-02801-f007:**
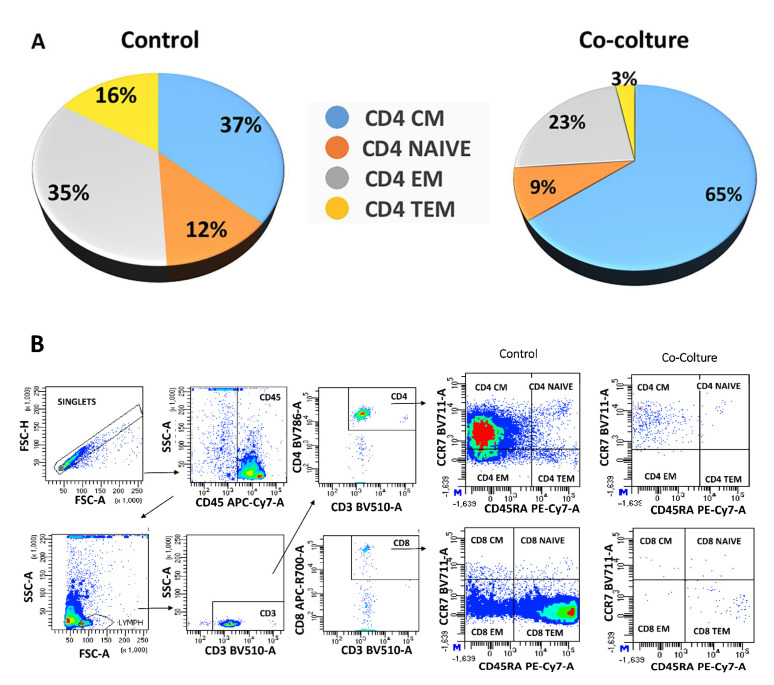
CD4^+^ T cell differentiation. Representative image of differentiation (**A**) and flow cytometry analysis (**B**) of CD4+ T lymphocyte population after 72 h of in vitro treatment. The percentages of the characteristic differentiating markers’ expression show how co-culture with hA-MSCs determines a decrease in the naïve CD4 component, with a consequent increase in the expression percentage of CD4 CM, CD4 EF, and CD4 TEMRA lymphocytes.

**Figure 8 ijms-25-02801-f008:**
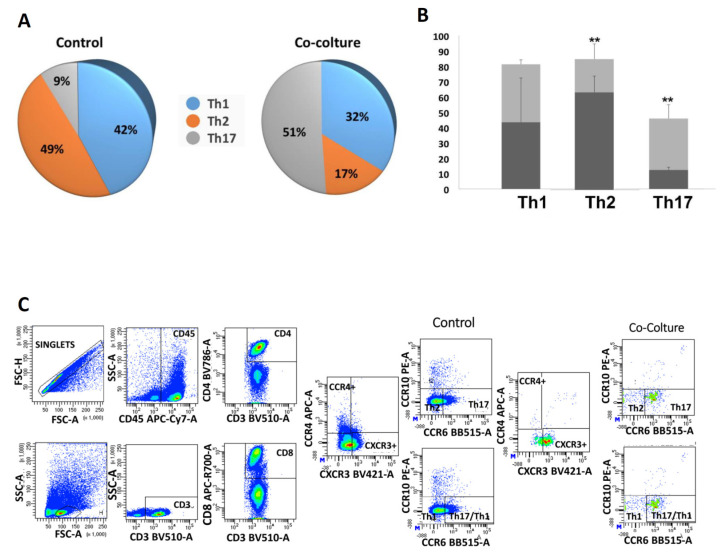
CD4+ T helper cell profiles. The representative image shows the trend of the T helper cell line. After 72 h of in vitro treatment, the presence of hA-MSCs determined an increase in the Th17 component compared to A-WBCs grown without co-cultured hA-MSCs (**A**). The graph (**B**) shows how hA-MSCs are able to reduce Th2 lymphocytic components (** *p*  <  0.01) and increase the Th17 lymphocytic components (** *p*  <  0.01) compared only to the A-WBCs’ control. Representative flow cytometry analysis (one out of four patients with cirrhosis) (**C**).

**Figure 9 ijms-25-02801-f009:**
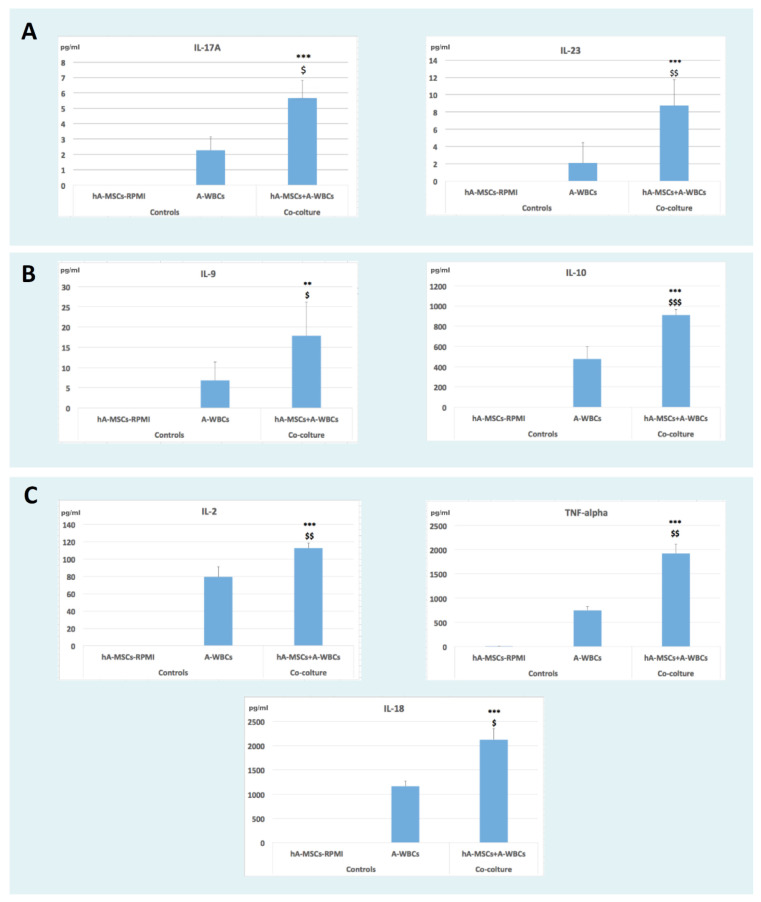
Cytokines. Graphs showing how the release of cytokines was influenced in ascites in the presence of hA-MSCs. Significant increases were seen for IL-17A and IL-23 (**A**); IL9 and IL-10 (**B**); and IL-2, TNF-α, and IL-18 (**C**) after 72 h of co-culture. Statistics were obtained by comparing co-culture with controls: hA-MSCs-RPMI (*) and A-WBCs ($). Values were statistically significant at *p* < 0.05 ($), *p* < 0.01 (**,$$), and *p* < 0.001 (***,$$$).Values are expressed as means of cytokines in pg/mL concentration.

**Table 1 ijms-25-02801-t001:** Features of Th cell subsets. The table shows a selection of surface markers and cytokines that can be used for a basic characterization via flow cytometry.

Th Cell Subset	Major Function	Surface Markers	Cytokines Secreted Involved in Regulation
TH1	Immunity against intracellular pathogens	CD4+CXCR3+	IFN-γ
TH2	Response to parasite infections	CD4+CCR4+CCR6–	IL-4, IL-5, IL-13
TH17	Response to fungi and extracellular bacteria	CD4+CCR4+CCR6+	IL-17A, IL-22

**Table 2 ijms-25-02801-t002:** Summary of clinical and biochemical data of enrolled patients with cirrhosis.

	Patient 1	Patient 2	Patient 3	Patient 4
Age/sex	55/M	62/M	50/M	71/M
Creatinine mg/dL	2	0.75	1	1.4
Total bilirubin mg/dL	24.4	3.7	3.7	3.1
AST/ALT U/L	82/17	54/36	29/17	21/14
Albumin gr/dL	2.5	2.8	2.7	2.8
Sodium mmol/L	137	139	136	134
Hemoglobin gr/dL	8.8	8.1	9.6	9.9
Platelets U/L	41.000	148.000	82.000	65.000
White blood count U/L	8430	7340	7890	6570
PMN (§) U/L	0.08	0.015	0.038	0.023
INR (#)	1.2	1.7	1.9	1.3
Ascites	Poorly controlled	Poorly controlled	Poorly controlled	Poorly controlled
Encephalopathy	None	None	None	None
Child–Pugh score	11	11	12	10
MELD score	28	18	20	20

§ Polymorphonuclear cell count (ascites). # International normalized ratio.

**Table 3 ijms-25-02801-t003:** Monoclonal antibodies used in flow cytometry.

Marker	Conjugation	Clone	Dilution	Manufacturer
CD3	FITC	HIT3a	1:20	BD Biosciences
CD4	PE-Cy7	SK3	1:20	Invitrogen (Waltham, MA, USA)
CD8	APC-Cy7	OKT8	1:20	Caprico Biotechnologies (Norcross, GA, USA)
CD14	APC-Cy7	MφP9	1:20	BD Biosciences
CD15	PerCP	HI98	1:20	BD Biosciences
CD16	PE	3G8	1:20	BD Biosciences
CD56	PE	MY31	1:20	BD Biosciences
CD45	PerCP	alone1	1:20	BD Biosciences

## Data Availability

The data presented in this study are available on request from the corresponding author.

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
