# Peer review of "Human Amniotic MSC Response in LPS-Stimulated Ascites from Patients with Cirrhosis: FOXO1 Gene and Th17 Activation in Enhanced Antibacterial Activation"

_ijms, 2024, doi:10.3390/ijms25052801_

Round 1
Reviewer 1 Report (New Reviewer)
Comments and Suggestions for Authors
I greatly appreciate the author's revision. I accept the manuscript in present form for publication
Author Response
We thank the reviewer for the positive statement.
Reviewer 2 Report (New Reviewer)
Comments and Suggestions for Authors
The authors described the upregulation of FOXO1, CXCL5, CXCL6, CCL20, and MAPK13 in hA-MSCs while promoting bacterial clearance, and a shift in the immune response towards a Th17 lymphocyte phenotype after 72 hours treatment. In an in vitro SBP model, they employed omics techniques (Next-generation Sequencing) to investigate how hA-MSCs modify the cross-talk between immune cells in LPS-stimulated ascitic fluid. They also validated data obtained by qRT-PCR, Citofluorimetric analysis, and Luminex assay. However, this reviewer has several concerns.
Comments
1. This study showed gene expression and immune response characterized by Th17 lymphocyte phenotype. However, the meaning and underlying mechanism of these changes in hA-MSCs response is unclear. Also, the clinical application of these findings should be discussed.
2. This reviewer wonders whether gene expression and lymphocyte phenotype are reproduced in other sample sets. Validation study needs to be taken into account.
3. There are many flaws as follows.
1) In Figure 1, A and B are utilized in the figure, but they are not mentioned in the main text and figure legend.
2) In Figure 3, what does this graph indicate? The units of the x-axis and y-axis should be described at least.
3) On lines 147-148, Figure 6 appears before Figure 4. It is strange.
4) On line 216, AF appears without an abbreviation. AF (Ascites fluid) should be added.
5) In Figure ï¼™, A, B, and C are utilized in figure legend, but not in figure and main text.
6) On line 368, MATERIAL E METHODS should be MATERIALS AND METHODS
Comments on the Quality of English LanguageThere are many flaws.
Author Response
The authors described the upregulation of FOXO1, CXCL5, CXCL6, CCL20, and MAPK13 in hA-MSCs while promoting bacterial clearance, and a shift in the immune response towards a Th17 lymphocyte phenotype after 72 hours treatment. In an in vitro SBP model, they employed omics techniques (Next-generation Sequencing) to investigate how hA-MSCs modify the cross-talk between immune cells in LPS-stimulated ascitic fluid. They also validated data obtained by qRT-PCR, Cytofluorimetric analysis, and Luminex assay. However, this reviewer has several concerns.
Comments
- This study showed gene expression and immune response characterized by Th17 lymphocyte phenotype. However, the meaning and underlying mechanism of these changes in hA-MSCs response is unclear. Also, the clinical application of these findings should be discussed.
We thank the reviewer for raising this point. The text has been modified in accordance with the requests.
- This reviewer wonders whether gene expression and lymphocyte phenotype are reproduced in other sample sets. Validation study needs to be taken into account.
The validation was conducted in four different ascites sample sets obtained from four cirrhotic patients. The NGS date was validated using different methods:
1) Real time-PCR, in order to confirm the upregulation of genes of interest;
2) Flow cytometric analysis, in order to evaluate the phenotype of CD4+ lymphocyte state obtained by NGS regarding the upregulation of Th17 pathways in the presence of hA-MSCs.
3) Luminex assay, in order to confirm both NGS and Flow cytometric analysis data through the validation of the cytokines responsible for the Th17 CD4+ pathway (IL-17 and IL-23).
- There are many flaws as follows.
- In Figure 1, A and B are utilized in the figure, but they are not mentioned in the main text and figure legend.
We thank the reviewer for this indication. We apologize for the mistake, and have modified the Figure as requested.
- In Figure 3, what does this graph indicate? The units of the x-axis and y-axis should be described at least.
The graph indicates gene expression distribution in hA-MSCs pools cultured alone and in co-culture, as described in the Legend. We thank the reviewer for the indication regarding the need to add the units of the x-axis and y-axis. We have modified the Figure as requested.
- On lines 147-148, Figure 6 appears before Figure 4. It is strange.
We thank the reviewer for pointing out this error. The text has been corrected by inserting Figure 4 and not 6.
- On line 216, AF appears without an abbreviation. AF (Ascites fluid) should be added.
We thank the reviewer for this observation. We have modified the text as requested.
- In Figure ï¼™, A, B, and C are utilized in figure legend, but not in figure and main text.
We thank the reviewer for pointing out this error. We have modified the Figure by adding A, B and C because they are mentioned in the Discussion.
6) On line 368, MATERIAL E METHODS should be MATERIALS AND METHODS
We thank the reviewer for the precise analysis; the text has been modified as requested.
(x) Extensive editing of English language required
The manuscript was revised by a professional language service, as requested.
Reviewer 3 Report (New Reviewer)
Comments and Suggestions for Authors
The authors present a very interesting study in which they demonstrate that human amnion-derived mesenchymal cells in the presence of in vitro SBP model, obtained through in vitro stimulation with LPS of ascitic fluid from decompensated cirrhotic patients, promotes a robust Th17 immune responses.
The figures are representative of the results presented in the text, perhaps it would be advisable to reduce the size of the letters in figure 3.
In Table 2, the INR is in monetary units (€), which should be corrected if it is an error.
In summary, the structure of the article is correct, the authors have made the suggested modifications and the article from my point of view is ready for publication in this journal, after correction of the details mentioned above.
Author Response
The authors present a very interesting study in which they demonstrate that human amnion-derived mesenchymal cells in the presence of in vitro SBP model, obtained through in vitro stimulation with LPS of ascitic fluid from decompensated cirrhotic patients, promotes a robust Th17 immune responses.
The figures are representative of the results presented in the text, perhaps it would be advisable to reduce the size of the letters in figure 3.
We thank the reviewer for noticing this. The figure has been modified, as requested.
In Table 2, the INR is in monetary units (€), which should be corrected if it is an error.
We thank the reviewer. The text has been modified, as requested.
In summary, the structure of the article is correct, the authors have made the suggested modifications and the article from my point of view is ready for publication in this journal, after correction of the details mentioned above.
Reviewer 4 Report (New Reviewer)
Comments and Suggestions for Authors
In this manuscript, Pampalone et al. provide insights into the mechanisms governing the role of the immune system in spontaneous bacterial peritonitis in human amnion mesenchymal stem cells (hA-MSCs) isolated from patients with liver cirrhosis and co-cultured with lymphocytes.
Although the study is potentially interesting, some issues should be addressed:
The title is confusing and too long. For example, the concepts of "2 FOXO1 Gene and Th17 Triggering," are rather inexact and confusing."hA-MSCs" should not appear as an acronym but as a full name.
Figure style is not homogeneous. In addition, some Figures have few data and can be merged: for example, Figure 3/4/5/6 might somehow be merged.
In general, it is not an easy to follow the study. Too many concepts appear and each paragraph describes an experiment without a concluding sentence. Providing concluding sentences all along the article will guide the reader to follow the story.
Comments on the Quality of English LanguageEnglish is not bad, but the organization and the writing style is not clear.
Author Response
In this manuscript, Pampalone et al. provide insights into the mechanisms governing the role of the immune system in spontaneous bacterial peritonitis in human amnion mesenchymal stem cells (hA-MSCs) isolated from patients with liver cirrhosis and co-cultured with lymphocytes.
Although the study is potentially interesting, some issues should be addressed:
The title is confusing and too long. For example, the concepts of "2 FOXO1 Gene and Th17 Triggering," are rather inexact and confusing."hA-MSCs" should not appear as an acronym but as a full name.
We thank the reviewer for pointing this out. The title had already been modified in the previous revision following the instructions of the Academic Editor who requested to specify the details. However, we followed the reviewer’s suggestion by trying to shorten it without altering the changes requested by the Editor who had accepted this new version of the title. We also removed the abbreviation hA-MSCs, as requested.
Figure style is not homogeneous. In addition, some Figures have few data and can be merged: for example, Figure 3/4/5/6 might somehow be merged.
We thank the reviewer for noticing this. We have modified the style in some figures to merge the style, but have left the Figure separated so as not to redesign the draft.
In general, it is not an easy to follow the study. Too many concepts appear and each paragraph describes an experiment without a concluding sentence. Providing concluding sentences all along the article will guide the reader to follow the story.
We thank the reviewer for the analysis. The text has been modified as requested. Furthermore, we have added an overall description of the study in the final part of the Discussion to direct the reader.
(x) Extensive editing of English language required
The manuscript has been revised by a professional language service, as requested.
Round 2
Reviewer 2 Report (New Reviewer)
Comments and Suggestions for Authors
Regarding comment#2, the author replied that the validation was conducted in four different ascites sample sets. However, no data was shown. How the validation was conducted is needed.
In addition, the Y-axis is not labeled in Figure 9.
Author Response
Regarding comment#2, the author replied that the validation was conducted in four different ascites sample sets. However, no data was shown. How the validation was conducted is needed.
In addition, the Y-axis is not labeled in Figure 9.
We pointed in the manuscript that the NGS data have been validated on the samples set of 4 patients by qRT- PCR but also by flow cytometry and Luminex, as indicated in the Results (lanes 201-204, 215-216, 238-243, -276, 286-287) and in Material and Methods (lanes 605-608, 629-630, 648-651, 661-663).
We thank the reviewer for the noticing the flaw in Figure 9 that was modified in the resubmitted version.
We hope that this new version will satisfy the requested points
Reviewer 4 Report (New Reviewer)
Comments and Suggestions for Authors
Issues have been addressed.
Thank you very much for the effort.
Author Response
We thank you for your positive support.
Round 3
Reviewer 2 Report (New Reviewer)
Comments and Suggestions for Authors
This reviewer understands that the authors confirmed NGS data by other methods. However, number of ascites samples is minimal. This reviewer thinks the limitation should be described in the discussion.
Author Response
This reviewer understands that the authors confirmed NGS data by other methods. However, number of ascites samples is minimal. This reviewer thinks the limitation should be described in the discussion.
We thank the reviewer for the indication. We modified the text as requested.
This manuscript is a resubmission of an earlier submission. The following is a list of the peer review reports and author responses from that submission.
Round 1
Reviewer 1 Report
Comments and Suggestions for Authors
The article addresses an original topic, brings something new regarding the management of spontaneous bacterial peritonitis, but it is not specified what type of study was conducted.
The title reflects the novelty element of the study, is concise and mention the research directions of the study.
The structure of this article does not correspond to the structure of the abstract, respectively introduction, materials and methods, results, discussions, and conclusions; please review the order of items.
The criteria of inclusion in the study and the population included in the study are well specified, but it is not clearly specified whether the informed consent was obtained from all patients involved.
Further research directions are discussed, the results and their statistical analysis are clearly expressed, as well as the method of conducting the study but the discussions do not mention the strengths and the weaknesses of the study.
Author Response
The article addresses an original topic, brings something new regarding the management of spontaneous bacterial peritonitis, but it is not specified what type of study was conducted.
We thank the reviewer for pointing that, the text was modified as requested
The title reflects the novelty element of the study, is concise and mention the research directions of the study.
We thank the reviewer for the statement about the title, however as requested from the editor we slightly modified it in accordance to the study. We hope this would not interfere with this judgment.
The structure of this article does not correspond to the structure of the abstract, respectively introduction, materials and methods, results, discussions, and conclusions; please review the order of items.
We thank the reviewer for noticing that, the abstract was modified as requested
The criteria of inclusion in the study and the population included in the study are well specified, but it is not clearly specified whether the informed consent was obtained from all patients involved.
Thank for your comment, as requested from journal policies we indicated it in 4.1 and 4.2 paragraphs and in the Informed Consent Statement paragraph: “Signed informed consent were obtained from patients enrolled in the study”
Further research directions are discussed, the results and their statistical analysis are clearly expressed, as well as the method of conducting the study but the discussions do not mention the strengths and the weaknesses of the study.
We thank the reviewer for its suggestions, the text was modified as requested
Reviewer 2 Report
Comments and Suggestions for Authors
FOXO1 gene and Th17 activation: key factors of hA-MSC Antibacterial action in Spontaneous Bacterial Peritonitis due to carbapenem-resistant Enterobacterales
The conventional method of ascitic fluid culture detects bacteria in only 50% of cirrhotic patients with neutrocytic ascites and suspected spontaneous bacterial peritonitis (SBP). In addition, the rise of antibiotic resistance requires alternative therapeutic strategies.
In this study, it was employed omics techniques (Next-generation Sequencing) to investigate the mechanisms by which hA-26 MSCs modify the dialogue between immune cells in the ascitic fluid infected by carbapenem resistant Enterobacterales. Data obtained were validated by qRT-PCR, Citofluorimetric analysis as well as Luminex assay. Results highlight the upregulation of certain transcription factors such as FOXO1, CXCL5, CXCL6, CCL20, 29 and MAPK13 in hA-MSCs while promoting bacterial clearance, and a shift in the immune response towards a Th17 lymphocyte phenotype after 72 hours treatment. Outcomes provide support to use hA-MSCs for the prevention and treatment of SBP, offering a possible alternative to antibiotic therapy. References must be up-dated. Research design is appropriate. Research design is appropriate. Results are clearly presented and conclusions are supported by the results.
Author Response
FOXO1 gene and Th17 activation: key factors of hA-MSC Antibacterial action in Spontaneous Bacterial Peritonitis due to carbapenem-resistant Enterobacterales
The conventional method of ascitic fluid culture detects bacteria in only 50% of cirrhotic patients with neutrocytic ascites and suspected spontaneous bacterial peritonitis (SBP). In addition, the rise of antibiotic resistance requires alternative therapeutic strategies.
In this study, it was employed omics techniques (Next-generation Sequencing) to investigate the mechanisms by which hA-26 MSCs modify the dialogue between immune cells in the ascitic fluid infected by carbapenem resistant Enterobacterales. Data obtained were validated by qRT-PCR, Citofluorimetric analysis as well as Luminex assay. Results highlight the upregulation of certain transcription factors such as FOXO1, CXCL5, CXCL6, CCL20, 29 and MAPK13 in hA-MSCs while promoting bacterial clearance, and a shift in the immune response towards a Th17 lymphocyte phenotype after 72 hours treatment. Outcomes provide support to use hA-MSCs for the prevention and treatment of SBP, offering a possible alternative to antibiotic therapy. References must be up-dated. Research design is appropriate. Research design is appropriate. Results are clearly presented and conclusions are supported by the results.
We thank the reviewer for its positive statements about our work. We check and modified the text in accordance with its and other reviewer suggestions.